# *Bacillus subtilis* Spores as a Vaccine Delivery Platform: A Tool for Resilient Health Defense in Low- and Middle-Income Countries

**DOI:** 10.3390/vaccines13100995

**Published:** 2025-09-23

**Authors:** Atiqah Hazan, Hai Yen Lee, Vunjia Tiong, Sazaly AbuBakar

**Affiliations:** 1Tropical Infectious Diseases Research & Education Centre (TIDREC), Universiti Malaya, Kuala Lumpur 50603, Malaysia; atiqahazan@um.edu.my (A.H.); leehaiyen@um.edu.my (H.Y.L.); evationg@um.edu.my (V.T.); 2Institute for Advanced Studies (IAS), Universiti Malaya, Kuala Lumpur 50603, Malaysia

**Keywords:** *Bacillus subtilis*, spores, vaccine, pandemic, health defense, infectious diseases

## Abstract

The COVID-19 pandemic exposed the urgent need for innovative tools to strengthen pandemic preparedness and health defense, especially in low- and middle-income countries (LMICs). While vaccination has been the cornerstone of the defense strategy against many infectious agents, there is a critical gap in vaccine equity, ensuring it is accessible to all, especially among the most vulnerable populations. The conventional vaccine delivery platforms, through parenteral administration, face notable limitations, including reliance on trained personnel, sterile conditions, and cold chain logistics. The parenteral vaccines often fail to induce robust mucosal immunity, which is critical for preventing infections at mucosal surfaces, the primary entry point for many pathogens. *Bacillus subtilis*, a Gram-positive, spore-forming bacterium, has emerged as a promising platform for mucosal vaccine delivery owing to its Generally Recognized as Safe (GRAS) status. Its robust spores are highly resilient to harsh environmental conditions, which may eliminate the need for cold chain storage and further facilitate distribution in LMICs. This review explores the potential of *B. subtilis* as a next-generation vaccine delivery platform, focusing on its unique characteristics, mechanisms of action, and applications in addressing global health challenges. This review also examines existing research demonstrating the safety, immunogenicity, and efficacy of *B. subtilis* spore-based vaccines while identifying limitations and future directions for optimization as a scalable and adaptable solution for resilient health defense, particularly in LMICs.

## 1. Introduction

The COVID-19 pandemic exposed critical weaknesses in our capacity to respond to large-scale health emergencies caused by rapidly spreading pathogens [1]. Beyond the devastating toll on mortality and morbidity, it severely disrupted livelihoods, education systems, and healthcare services, with low- and middle-income countries (LMICs) disproportionately affected due to constrained resources and vulnerable health systems [1,2,3,4,5]. Vaccination has remained a cornerstone in controlling the pandemic [6,7,8,9], reportedly saving an estimated 20 million lives during the peak of the rollout [10]. This remarkable achievement, however, was unevenly realized across the globe as only about 2% of vaccines reached African communities, and many countries were ultimately unable to secure their allocated doses [11]. This stark disparity in vaccine access posed a significant challenge to future global pandemic preparedness. This strongly emphasizes the importance of building a more equitable, sustainable, and resilient health defense system to mitigate future pandemic threats. Importantly, resilient health defense extends beyond pandemic preparedness by focusing on long-term, ongoing strategies to expand access to healthcare, reduce health inequities, and integrate innovations in response and recovery during health emergencies.

Addressing these challenges requires the development of innovative vaccine platform technologies that can strengthen resilient health defense systems. Such platforms enable the rapid production of vaccines against diverse pathogens using a flexible, universal backbone. Among these, mRNA and viral vector vaccines have emerged as some of the most advanced and widely used technologies [12]. These vaccine technologies came to the fore during the COVID-19 pandemic and have been credited with being effective in preventing symptomatic and severe cases [13,14,15]. Despite this success, several limitations have been associated with the usage of these vaccines. Foremost among these, the mRNA and viral vector vaccines were administered intramuscularly [16]. These methods often require trained personnel for administration, sterile conditions, and specialized equipment, making them less feasible in resource-limited settings. Administration can also result in localized reactogenicity at injection sites and, in rare cases, more serious adverse events, contributing to vaccine hesitancy [17,18,19]. Reported serious adverse events include anaphylaxis, myocarditis, thrombosis, and thrombocytopenia [20,21]. Another major limitation is their restricted ability to induce robust mucosal immunity [22], which is critical for preventing infections at mucosal surfaces, the primary entry point for many pathogens [23,24,25]. This limitation is particularly critical in the case of respiratory and gastrointestinal diseases, thus highlighting the need for innovative approaches to vaccine delivery, especially for those targeting the stimulation of mucosal immunity.

*Bacillus subtilis*, a Gram-positive, spore-forming bacterium recognized by the Food and Drug Administration (FDA) as Generally Recognized as Safe (GRAS), has been extensively studied for many biotechnological and biomedical applications [26,27]. Due to its well-established safety profile, *B. subtilis* has been utilized for protein display/expression platforms and was demonstrated to perform as well as *Escherichia coli* in expressing proteins [28]. While *E. coli* has been a longstanding platform for recombinant protein expression, challenges such as the formation of inclusion bodies where the protein aggregates would require labor-intensive solubilization and refolding steps to recover functional proteins [29]. This would reduce yield and complicate downstream processing.

Unlike *E. coli*, a defining characteristic of *B. subtilis* is its ability to sporulate under nutrient-limiting conditions, producing spores capable of withstanding extreme heat, acidity, desiccation, and chemical exposure [30,31]. These spores can also germinate when the environment becomes favorable [26,27,30], hence making *B. subtilis* spores not only environmentally stable but also biologically active. They may also serve as a natural adjuvant through the activation of the MyD88-dependent innate signaling pathway, thereby promoting antigen presentation and immune activation [32,33,34]. This innate immune-stimulating property of *B. subtilis* spores eliminates the need for additional adjuvants, which are often required in most recombinant protein vaccine platforms [35,36]. The inherent stability and immunostimulatory properties of *B. subtilis* spores make them particularly attractive as a vaccine delivery platform, capable of presenting heterologous antigenic proteins. Spore-based vaccines can be administered via mucosal routes, thus inducing both systemic and mucosal immune responses [26,37].

The potential of spore display was first demonstrated with tetanus toxin fragment C (TTFC) on the surface of *B. subtilis* spores using CotB as the anchoring motif [38]. Since then, it has been utilized to express many other heterologous proteins for various applications, including as a vaccine delivery platform. This strategy has been shown to overcome several limitations of conventional vaccine platforms, such as the need for cold chain storage, high production costs, and the inability to effectively stimulate mucosal immunity. The advantages of spore-based vaccines position *B. subtilis* as a promising system for developing novel vaccine delivery strategies, particularly against infectious diseases where mucosal immunity is critical.

This review aims to highlight *B. subtilis* spores as a viable vaccine delivery platform, emphasizing its unique characteristics, mechanisms of action, and potential applications in addressing global health challenges, including pandemic preparedness, especially for LMICs. Specifically, the review focuses on research that demonstrates proof of concept for *B. subtilis* in delivering antigens as vaccines, with particular attention to its immunogenicity, safety, and efficacy. The selection criteria prioritize research with substantial evidence of success while discussing limitations and future directions to optimize the use of *B. subtilis* as a novel vaccine delivery platform.

## 2. Lessons Learned from the COVID-19 Pandemic

One of the most critical lessons from the COVID-19 pandemic is the importance of rapid vaccine development and deployment in controlling outbreaks [39,40,41,42]. The development of COVID-19 vaccines, such as mRNA-based and viral vector vaccines, within a year demonstrated the potential of platform technologies and the benefits of pre-emptive investments in biotechnology [43,44]. The pandemic highlighted the need for next-generation vaccines capable of providing broad and durable protection, enhancing mucosal immunity, and reducing transmission to maintain population-level immunity and improve preparedness for future outbreaks. Vaccines that are easily deployable, low-cost, low-risk, and effective in boosting immunity are particularly essential for LMICs.

It has been well documented that the COVID-19 pandemic also caused severe disruptions to routine vaccination campaigns for numerous vaccine-preventable diseases, including measles, diphtheria, polio, tetanus, meningitis, and pertussis, particularly among children worldwide [45,46,47,48]. According to reports from the World Health Organization (WHO), UNICEF, and Gavi, the Vaccine Alliance, at least 80 million children were at risk of contracting these diseases due to repercussions from COVID-19 pandemic-related containment challenges. [46]. Measures such as lockdowns and movement restrictions, implemented to curb the spread of COVID-19, significantly delayed the otherwise routine vaccination programs [49]. Additionally, global health priorities shifted towards mass vaccination against SARS-CoV-2, further contributing to the delays [48,50]. At the same time, the healthcare system faced significant challenges as many healthcare workers were redeployed to support COVID-19 response efforts, exacerbating the burden on an already overstretched sector. This reallocation of resources further disrupted routine vaccination services and amplified the challenges faced by healthcare providers, particularly in LMICs. Furthermore, SARS-CoV-2 infection and mortality rates were disproportionately high among healthcare workers, leading to staffing shortages that further strained healthcare delivery and hindered the provision of essential services, including routine vaccinations [51,52,53]. In light of these challenges, alternative vaccine platforms, such as oral or intranasal vaccines, have been proposed as potentially advantageous. These routes reduce reliance on trained personnel for injections, as demonstrated by the success of oral polio vaccine campaigns, which enabled large-scale administration with minimal infrastructure [54]. In addition, framing spore-based vaccines as pain-free alternatives to injections could improve public acceptance, particularly among vaccine-hesitant populations. While not a direct solution to the staffing shortages observed during the pandemic, such innovations may ease healthcare system pressures and enhance resilience by supporting more accessible and sustainable immunization strategies.

The COVID-19 pandemic experience emphasized how essential resilient vaccine manufacturing and supply chains are in ensuring vaccine equity. Despite unprecedented levels of vaccine production, significant disparities in distribution exposed the urgent need for more equitable mechanisms [55,56]. Initiatives by the Coalition for Epidemic Preparedness Innovations (CEPI), Gavi, the Vaccine Alliance, and the WHO, like COVAX, while promising in their goals, fell short of ensuring timely vaccine access for LMICs [57]. Reports revealed that many high-income countries stockpiled vaccines in the spirit of vaccine nationalism to secure booster doses, while dozens of LMICs struggled to administer even the first dose to their populations [57,58]. High-income countries also gained access to COVID-19 vaccines almost immediately after their rollout in December 2020, whereas LMICs had to wait until the first quarter of 2021 to begin their vaccination campaigns [59]. This delay was exacerbated by logistical challenges, limited supply, and inadequate infrastructure in LMICs, widening the gap in global immunization coverage [60]. To address such inequities in future pandemics, preparedness strategies must focus on diversifying global vaccine manufacturing sites and strengthening local production capacities, including in LMICs. Establishing regional manufacturing hubs and investing in local infrastructure can reduce reliance on centralized supply chains and improve the speed and equity of vaccine distribution. These measures would enhance self-reliance in LMICs and contribute to more robust and inclusive global health security.

The development of manufacturing facilities represents a significant challenge regarding the financial burden and the technical complexities involved in vaccine production [61]. The costs associated with establishing state-of-the-art facilities, meeting stringent regulatory standards, and ensuring a reliable supply chain can be prohibitive for LMICs. The cost of developing large-scale, product-specific vaccine manufacturing facilities was estimated to range from USD 500–700 million and usually takes approximately 7 years to be ready for the first batch manufacturing process [62,63,64]. However, adopting versatile vaccine platforms offers a potential solution to mitigate these challenges. Platform technologies, such as mRNA, viral vectors, or recombinant protein systems, enable rapid vaccine development and scaling up against emerging pathogens by utilizing standardized processes and infrastructure [64]. It was reported that the manufacturing cost of these vaccines was reduced by more than 20-fold in comparison to the conventional inactivated vaccine manufacturing [64]. These platforms, hence, can significantly reduce production timelines and costs by allowing manufacturers to pivot quickly from one vaccine target to another without overhauling the entire production process [64,65,66]. Moreover, platform-based vaccines often require less specialized manufacturing equipment due to the fixed platform base and standardized upstream and downstream processes, making it easier to establish modular facilities adapted for various vaccines [65,66]. This approach not only enhances pandemic preparedness but also promotes equitable access by empowering LMICs to develop and produce vaccines locally with reduced capital and operational costs, hence making cheaper vaccines accessible to the needy populations.

The COVID-19 pandemic highlighted significant logistical challenges in vaccine distribution, particularly the cold chain requirements for specific formulations such as for the mRNA vaccine. Regions with limited infrastructure encountered substantial barriers in deploying vaccines like mRNA formulations, which require ultracold storage conditions. For instance, the COVAX initiative, which invested nearly USD 2.4 billion to accelerate vaccine research and development, faced limitations in contracting mRNA vaccines due to the insurmountable challenges of delivering these vaccines to LMICs [67]. Despite these intense efforts, only three candidate vaccines, AstraZeneca/Oxford’s viral vector vaccine and two mRNA vaccines by Moderna and Pfizer/BioNTech were approved for emergency use during the pandemic [68]. However, the deployment of these vaccines in LMICs was hindered by inadequate cold chain infrastructure. According to the WHO, approximately 50% of vaccines are wasted globally each year due to logistical complications and cold chain failures [69]. Addressing these challenges requires a two-pronged approach. First, investing in developing thermostable vaccines that can remain effective at standard refrigeration or ambient temperatures is crucial for simplifying distribution and reducing wastage. Second, strengthening global cold chain systems, particularly in resource-limited settings, is vital to ensure equitable access to vaccines during future outbreaks. The former seems more cost-effective and better suited to being established in LMICs.

Simultaneously, vaccine hesitancy and the spread of misinformation emerged as significant barriers to achieving widespread immunization coverage [70]. This challenge was particularly pronounced with the use of mRNA technology in the COVID-19 vaccine development. Concerns surrounding vaccine quality and safety, along with the rapid development and deployment timelines, fueled public skepticism [71]. To address these challenges, transparent communication strategies and proactive efforts to counter misinformation are crucial for building public trust in vaccines and ensuring higher uptake during future health crises. For spore-based vaccine platforms, practical strategies include emphasizing the established safety of *B. subtilis* as a GRAS organism and highlighting the stability of spores without cold chain requirements. Clear messaging around needle-free mucosal delivery could also increase acceptance among hesitant groups. In addition, involving local healthcare providers and community leaders as advocates, presenting the platform as cost-effective for resource-limited settings, and underscoring its natural adjuvant properties can further strengthen public trust.

## 3. Methodology

The literature review was conducted using a systematic search strategy. Pertinent keywords included “*Bacillus subtilis*,” “surface display,” “spore display,” “recombinant,” “vaccine,” “antigen,” “immune response,” and “mucosal immunity. Boolean operators (AND, OR, NOT) were applied to refine the searches across scholarly databases such as PubMed, Scopus, and Web of Science. Only original experimental studies reporting in vitro or in vivo immunological data were included. Reviews, conference abstracts, purely in silico studies, and studies focused exclusively on veterinary pathogens or animal diseases were excluded, unless they provided broadly relevant methodological insights into *B. subtilis* vaccine development. After screening, the selected literature was organized thematically to highlight advances in antigen attachment strategies, immunogenicity studies, and manufacturing considerations for *B. subtilis* spore-based vaccines. The findings of the selected literature are organized in Table 1, which summarizes the pathogen targeted, antigen, mechanism of antigen carrier, strain, delivery route, and application for each study.

## 4. Biological Characteristics of Bacillus subtilis and Its Potential as a Vaccine Delivery Platform

### 4.1. Biological Characteristics of B. subtilis and Its Sporulation Cycle

*B. subtilis* is a well-characterized, extensively studied bacterium widely recognized for its long history of safe use in industrial and pharmaceutical applications. It is abundant in nature and can be isolated from many sources, including soil [113], marine environment [114], and terrestrial sources [115,116]. A key feature of *B. subtilis* is its ability to form endospores in nutrient-deprived environments, which are highly resilient and enable the bacterium to survive under harsh environmental conditions, including extreme heat, desiccation, and chemical stress [117]. It can remain dormant for a long time and can germinate into a vegetative state when the conditions become favorable [26]. *B. subtilis* grows under aerobic conditions and uses oxygen as the electron acceptor for its metabolic activity [118,119,120]. However, under an oxygen-deprived environment, *B. subtilis* may utilize an anaerobic metabolic pathway using either nitrate or nitrite as the final electron acceptor, or it may undergo sporulation in the absence of electron acceptors [119,120]. This allows for the *B. subtilis* to survive in the mammalian gastrointestinal (GI) tract by employing the anaerobic metabolic pathway and complete its entire life cycle from sporulation to germination and proliferation [121,122].

The sporulation cycle of *B. subtilis* is initiated in response to nutrient depletion and environmental stress (Figure 1). The process begins with activating the master regulator *Spo0A* in response to the activation of histidine sensor kinases (KinA, KinB, and KinC), which, upon phosphorylation (*Spo0A-P*), stimulates the expression of sporulation-specific genes [123]. Subsequently, the cell elongates and undergoes asymmetric division, generating a smaller forespore and a larger mother cell, separated by a polar septum [117]. The forespore is then engulfed by the mother cell membrane, resulting in a double-membraned structure [124]. Within this protected environment, the forespore synthesizes a specialized peptidoglycan layer known as the cortex, which contributes to its mechanical resilience [124]. Concurrently, the mother cell assembles multiple protective protein layers around the forespore, forming a robust spore coat [125]. Maturation of the spore involves critical biochemical adaptations, including accumulating dipicolinic acid (DPA) and calcium ions, stabilizing the DNA, and enhancing resistance to heat and desiccation [117]. Ultimately, the mother cell undergoes programmed lysis, releasing the mature, dormant spore into the environment, equipped to endure extreme conditions and germinate under favorable circumstances [117].

### 4.2. Laboratory Strains and Engineering for Antigen Delivery

Several laboratory strains of *B. subtilis* have been widely used to explore its biology and biotechnological applications. The most common *B. subtilis* strains adopted for antigen delivery platforms include 168, PY79, WB800N, and WB600 (Table 1). *B. subtilis* 168 was derived from *B. subtilis* Marburg and isolated as a Trp− mutant during the early days of the postwar era [126]. It has undergone laboratory adaptation with at least three DNA polymorphisms, one of which was the single-base duplication that resulted in the inactivation of *sfp* and *swrA*, which are responsible for the loss of swarming and multicellularity, distinguishing the strain from wild-type Marburg [127]. The X-ray mutation of the 168 resulted in the nine-base duplication that inactivated *gudB*, causing an increase in the growth rate of the *B. subtilis* in a glucose-ammonia minimal medium [126,127]. The PY79 strain is a phototroph, originating from the 168 and W23 strains, with the absence of the auxotrophic markers (*glnA100* and *metB5*) through two cycles of PBS1-mediated transduction. The PY79 strain has been extensively used in many -omics studies [128,129,130]. WB800N is a derivative of the 168 strain that has undergone genetic engineering to remove the extracellular proteases NprE, AprE, Epr, Bpr, Mpr, NprB, Vpr, and WprA [131]. This allows for a more stable expression of recombinant proteins. Similarly, W600 is also derived from the 168 strain with six of the protease genes, including *nprE*, *aprE*, *epr*, *mpr*, *bpr*, and *nprB*, leaving only 0.32% extracellular protease activity [132]. Such strain diversity provides versatile tools for molecular engineering and vaccine platform development.

The spore coat, assembled during sporulation, is of particular interest for vaccine applications because it provides natural scaffolds for antigen display. CotB was the first spore coat protein used for the surface display of exogenous proteins. It has a strongly hydrophilic C-terminal region composed of serine-rich repeats. This C-terminal region facilitates the fusion and surface localization of heterologous proteins. The expression of CotB is regulated by maternal cell-specific sigma factors and transcription factors like GerE and GerR [133]. Its assembly requires CotG and CotH, where CotG interacts directly with CotB to aid in its stability, while CotH protects CotG from proteolytic degradation [89,134]. Mutations in the cotB, cotG, or cotH genes were reported to disrupt the surface display of heterologous proteins [125]. CotC is a highly abundant coat protein that assembles in various forms, including a monomer of 12 kDa and a homodimer of 21 kDa [135]. CotC is regulated by CotH and CotE, which ensure its stability during assembly [135]. Overexpression of CotH allows CotC to accumulate in the mother cell before assembly into the spore coat [136]. CotG is another spore coat protein used for surface display, regulated by σ^ĸ^ and GerE [133]. It assembles as 32 and 36 kDa variants, with CotH playing a critical role in protecting CotG from proteolysis during sporulation [137,138]. Other spore coat proteins include CotZ, which is a crust protein, regulated by σ^E^, σ^ĸ^, and GerE [117]. Similarly, CgeA, another crust protein, is also controlled by σĸ and GerE for expression [139]. Each of these proteins has been shown to effectively present recombinant proteins on the spore surfaces or incorporate them within the spore itself. The choice of anchor protein used depends on the specific application and desired properties of the recombinant protein. However, the minimum requirement should be that the coat proteins have strong anchoring domains (146), resistance to protease hydrolysis (147), and compatibility with the intended exogenous proteins [140].

### 4.3. Advantages of B. subtilis as a Vaccine Platform

*Bacillus* species, particularly *B. subtilis*, have been extensively used as probiotics for human and animal consumption owing to their robust safety profile [27,141]. In many parts of Asia and Africa, *Bacillus* species are traditionally consumed as part of food delicacies, with no significant safety concerns reported over time [141,142,143]. Some notable examples are the Japanese dish, *Natto*, and the traditional Korean delicacy, *Cheonggukjang*, which utilize *B. subtilis* during fermentation for commercial production [141,144,145,146]. Beyond traditional foods, *Bacillus* spores are commonly employed as dietary supplements and functional foods due to their numerous health benefits, including gut health improvement, immune modulation, and antimicrobial properties [104,147,148,149]. This long history of safe use and the FDA’s GRAS status highlight the potential of *B. subtilis* as a viable platform for vaccine delivery and other therapeutic applications.

One of the primary advantages of *B. subtilis* spores is their remarkable stability under non-refrigerated conditions. In LMICs, where advanced manufacturing facilities and cold chain logistics are often limited, the *B. subtilis* spore-based vaccine offers an accessible and practical solution. The spores’ natural resilience to extreme environmental conditions allows them to be stored and transported at ambient temperatures, bypassing the cold chain infrastructure that most other vaccine platforms require. The spores of *B. subtilis* are naturally resilient to extreme environmental conditions, including high temperatures and desiccation. As a result, they can be stored and transported at ambient temperatures for extended periods, reducing the need for costly and complex refrigeration infrastructure. This is particularly advantageous in remote or underserved areas, where access to reliable cold chain systems can be limited or non-existent. By eliminating the cold chain requirement, *B. subtilis* spore-based vaccines can be distributed more efficiently, reaching populations in even the most challenging logistical environments. It was suggested that the recombinant spores remained immunogenic and protective even after an extended period of storage in high-temperature environments [79]. This stability facilitates the storage and transport of the *B. subtilis* spore-based vaccines, reducing the need for costly and complex cold chain systems. Upon exposure to favorable environmental conditions, such as the presence of water and nutrients, *B. subtilis* spores can rapidly germinate and resume metabolic activity [150]. This characteristic is crucial for the successful delivery of vaccines that depend on the germination of the *Bacillus* spores, as the spores can transition from a dormant to an active state upon entering the gastrointestinal tract.

*B. subtilis* possesses a relatively simple genome that is highly amenable to genetic manipulation [26]. This allows for the incorporation of foreign antigens, enabling the bacterium to serve as a carrier for targeted immune responses. Genetic modification of *B. subtilis* is straightforward, utilizing well-established molecular biology techniques to integrate foreign genes into its genome. It has been shown that the antigen presented on the spores remains stable in its native conformation [80]. Advanced approaches such as reverse vaccinology further enhance this adaptability [37]. Researchers can rapidly synthesize and express these antigens by screening the genomes of microorganisms to identify protective antigens that can be displayed on *B. subtilis* spores. This flexibility means that once a *B. subtilis* spore-based platform is developed and approved, it can be quickly adapted for other pathogens, significantly reducing development timelines during future pandemics.

*B. subtilis* spores have the potential to stimulate strong mucosal immunity when administered via mucosal routes such as oral, nasal, sublingual, and suppository. Mucosal immunity is critical for combating respiratory and gastrointestinal pathogens, as it represents the first line of defense at the primary sites of infection. By stimulating immune responses directly at the mucosal surfaces of the respiratory and gastrointestinal tracts, *B. subtilis* spore-based vaccines can provide rapid and localized protection against pathogens such as SARS-CoV-2 and influenza. This mucosal immunity is particularly important in the context of respiratory viruses, as it may help prevent initial infection and reduce transmission rates. Moreover, mucosal delivery eliminates the need for needle-based administration, enabling self-administration and increasing accessibility in resource-limited settings.

### 4.4. Immunological Properties of B. subtilis Spores

The resilient nature of *B. subtilis* spores also allows them to withstand the stomach’s acidic environment and the concentration of bile salts in the duodenum, enabling the spores to survive the harsh conditions of gastric transit [151]. *B. subtilis* spores have been shown to interact with the gut-associated immune microenvironment, influencing the development and function of gut-associated lymphoid tissue (GALT) [152,153,154]. GALT is central in initiating and regulating immune responses, particularly mucosal immunity. Upon reaching the intestines intact, the spores can interact with the gut immune system through several mechanisms. They can survive uptake by phagocytic cells, such as M cells, in the intestinal mucosa and be transported to Peyer’s patches [83]. Within these cells, *B. subtilis* spores may germinate in the phagolysosome, facilitating their interaction with antigen-presenting cells (APCs) [73,78,151] before transiting to the efferent lymph nodes [83]. After germination in the upper intestines [151], *B. subtilis* can re-sporulate in the lower intestines, providing prolonged exposure to the immune system [155]. It was also suggested that germination resulted in spore coat shedding, and the digestion of the spore coat debris facilitates antigen presentation [103]. However, the majority of spores are excreted without germinating [83]. Notably, recombinant spores have also been reported to stimulate immune responses without requiring germination. This indicates that multiple pathways exist for spores to activate host immunity, serving as an initial trigger for downstream adaptive responses [81].

The immunostimulatory effect of *B. subtilis* spores is mediated through the MyD88-dependent signaling pathway, a key adaptor molecule used by most Toll-like receptors (TLRs) such as TLR-2 and TLR-4 [32,33,34]. Specifically, the adjuvanticity of *B. subtilis* spores is attributed to recognition by TLR2, as evidenced by the absence of an adjuvant effect in TLR2 knockout mice [34]. The bacterium’s natural adjuvanticity, mediated through its interactions with innate immune receptors, eliminates the need for external adjuvants, making it an even more efficient vaccine delivery platform [36,156]. This may obviate the need to use a vaccine preparation consisting of >90% recombinant spores. *B. subtilis* spores have been shown to induce dendritic cell (DC) maturation, a critical component of the adaptive immune system (Figure 2). This process includes increased expression of pro-inflammatory cytokines and MHC-I, MHC-II, and CD40 molecules [34]. Inactivated spores were shown to induce immune activation effectively [32,157]. However, live spores are more effective in inducing MHC-II surface expression and IL-12 secretion, likely due to spore germination occurring either in the culture medium or inside the DCs after phagocytosis [34]. Significantly, inactivated spores can induce immune responses as effectively as live spores, providing a safer alternative without compromising immunogenicity [35,157]. This characteristic promotes the use of inactivated spores as a natural adjuvant, particularly in settings where live spores might raise biosafety concerns. Furthermore, *B. subtilis* spores stimulate the upregulation of additional immune-related molecules, such as CCR7, PD-L1, and PD-L2, which are crucial for inflammatory responses and further enhance the stimulation of immune pathways [33].

Once activated, dendritic cells migrate from Peyer’s patches and other mucosal inductive sites to mesenteric lymph nodes, where they present spore-derived antigens to naïve T cells. This process drives CD4^+^ T-cell differentiation into Th1, Th2, Th17, and Treg subsets, alongside CD8^+^ cytotoxic responses. Concurrently, B cells are activated to produce plasma cells secreting antigen-specific antibodies. At mucosal sites, this primarily results in secretory IgA production, whereas systemically it supports serum IgG responses and antigen-specific T-cell responses. Mucosal inductive sites such as GALT, nasal-associated lymphoid tissue (NALT), and bronchial-associated lymphoid tissue (BALT) contribute to these responses by supporting local IgA production and the establishment of tissue-resident memory T cells (TRM). These TRM cells provide long-lasting site-specific protection, complementing systemic responses mediated by circulating IgG and effector T cells. Through this dual action, spore-based vaccines can establish both frontline mucosal defense and durable systemic immunity (Figure 2).

Several studies provide evidence that recombinant *B. subtilis* spores can elicit strong adaptive immune responses across different routes of administration. In a murine malaria model, nasal administration of spores expressing *Plasmodium falciparum* rPfCSP induced a significant increase in antigen-specific IgG levels compared to controls [111]. Similarly, oral or intranasal immunization with *Clostridium tetani* CotB-TTFC spores induced detectable TTFC-specific secretory IgA in fecal samples, with peak titers observed at day 33 [83]. Parallel increases in TTFC-specific serum IgG were observed from day 29–33 onwards, indicating the capacity of spore-based vaccines to induce both systemic and mucosal humoral immunity [83]. Recombinant spores expressing *Mycobacterium tuberculosis* Ag85B-CFP10 secretory antigens elicited strong IgG responses against this antigen, whereas BCG-immunized mice showed minimal antibody reactivity, consistent with the absence of CFP10 in BCG [92]. Recombinant spores expressing SARS-CoV-2 spike antigens used as intranasal boosts after systemic priming (with rSWuh protein or AZD1222) markedly enhanced mucosal immunity [32]. Salivary antigen-specific IgA was significantly elevated (*p* < 0.001) compared to priming alone, while lung IgA extracts from boosted mice showed titers approximately two-fold higher than those of groups receiving wild-type spores [32]. Serum IgG responses were also significantly higher (*p* < 0.01) in boosted groups, confirming the ability of spores to strengthen both mucosal and systemic immunity in heterologous prime–boost regimens [32].

In addition to humoral responses, recombinant spores also elicit cellular immunity. Splenocytes from mice immunized with spores expressing SARS-CoV-2 antigens secreted higher levels of IL-2, TNF-α, and IL-5 upon antigen restimulation, indicating activation of cytotoxic and helper T cell subsets [32]. Similarly, spores engineered with *Clostridium difficile* FliD protein induced strong FliD-specific IgG and mucosal IgA, along with elevated IL-2, IL-17A, and IFN-γ, while maintaining low IL-10 levels, consistent with a pro-inflammatory Th1/Th17-biased profile [77]. It was observed that splenocytes from mice immunized with recombinant spores expressing *Mycobacterium tuberculosis* Ag85B-CFP10 secretory antigens also exhibited a significantly higher number of Ag85B-specific IFN-γ-secreting cells compared to those from mice vaccinated with wild-type non-recombinant spores or from naïve controls [92].

Importantly, recombinant spores have also been shown to induce efficacious adaptive responses following parenteral administration. In one study, subcutaneous injection of *B. subtilis* expressing human papillomavirus (HPV) HPV33 L1 protein resulted in serum IgG titers against HPV33 L1 that were approximately 100-fold higher than in controls [99]. Cellular immunity was also demonstrated, with splenocytes from immunized mice producing IFN-γ at levels three-fold higher than mock-immunized animals [99].

Several preclinical challenge studies have demonstrated the capacity of *B. subtilis* spore-based vaccines to induce functional protective immunity across different pathogens. In one study, recombinant spores displaying the protective antigen (PA) of *Bacillus anthracis* conferred significant protection in mice challenged with lethal doses of *B. anthracis* spores. Mice immunized with PA–spore vaccines showed markedly higher survival compared with naïve controls (*p* < 0.01) [72]. The route of administration influenced efficacy, with intraperitoneal immunization offering the highest survival, followed by intranasal and then oral delivery, although all recombinant groups showed significant protection compared to controls [72].

Similar protection was observed in an influenza model using spores adsorbed with NIBRG-14 (H5N2) antigens [35]. Mice immunized intranasally with as little as 0.02 μg HA adsorbed onto spores achieved full protection against a lethal challenge with H5N2, whereas antigen alone provided only partial protection [35]. Even heat-killed spores administered without adsorbed antigen resulted in 60% protection, reflecting the innate adjuvanticity of the spore itself [35]. In another study, recombinant spores expressing the conserved influenza M2e-FP antigen provided robust cross-protection against H1N1 [101]. Intratracheal immunization with RSM2eFP spores provided 100% protection against a 20 × LD50 challenge of A/PR/8/34 (H1N1) virus and 80% protection at 40 × LD50, whereas intragastric delivery was less effective, again emphasizing the importance of route of administration [101].

Protective efficacy has also been demonstrated against parasitic infection. Oral vaccination of hamsters with recombinant spores expressing the extracellular loop (LEL) of the tetraspanin Ov-TSP-2 from *Opisthorchis viverrini* significantly reduced worm burdens compared to both CotC-expressing spores and non-spore controls (*p* < 0.0001) [110]. This partial but statistically significant protection indicates that spore-based vaccines may also be applied to helminth infections, including carcinogenic liver flukes of major public health importance. Taken together, these findings show that spore-based vaccines are capable of inducing protective immunity across bacterial, viral, and parasitic diseases. The findings also demonstrate the influence of delivery route and the combined effects of recombinant antigen display with the intrinsic adjuvanticity of spores in shaping immune protection.

## 5. Mechanisms of Vaccine Delivery: Recombinant Antigen Encapsulation and Non-Recombinant Surface Display

Many studies have demonstrated the feasibility of utilizing the *B. subtilis* spores to express heterologous antigens of various bacteria, viruses, and parasites for vaccine development (Table 1). *B. subtilis* can function as an effective vaccine carrier through two primary mechanisms: antigen encapsulation within the spores and antigen surface display [158,159]. Both approaches leverage the unique biological properties of *B. subtilis* spores to deliver heterologous antigens in a way that stimulates effective immune responses.

The recombinant expression of heterologous proteins involves encapsulation within the spore coat itself, providing a protective environment that shields the antigen from degradation during gastrointestinal transit [160]. This method ensures that the antigens remain intact and bioavailable as the spores travel through the gastrointestinal tract and stomach’s acidic environment. Early approaches involved genetically fusing foreign antigens to spore coat proteins. These coat proteins act as anchor points, ensuring the antigens are stably incorporated into the *B. subtilis* spore surface [159]. Using this approach, DNA fragments encoding the gene of interest and the spore coat protein are fused through the C- or N-terminal. These recombinant gene are then introduced into the *B. subtilis* via plasmids transformation, where they are induced for expression in the mother cell [160]. Upon the initiation of sporulation and gene expression, the recombinant proteins are incorporated into the growing spore structure, resulting in the assembly of spores displaying the heterologous proteins without affecting the structure and functions of the spores [159,160]. Several of the *B. subtilis* spore coat proteins have been studied and utilized as the recombinant protein anchor motifs. Notable examples of the spore coat proteins are CotB [73,75,76,78,81,88,96,97,101,102,103], CotC [32,73,78,79,87,89,90,92,94,103,104,105,106,107,108,110,112], CotG [76], CgeA [103], and CotZ [103].

It has been reported that the expression of heterologous antigens in *B. subtilis* can be achieved without relying on the use of anchor proteins [37,72,74,84,85,95,98,99,100]. In this approach, a modified promoter region is incorporated together with the antigen into the *B. subtilis* genome [37,72,85,94]. The *cry1Aa* and *cry3Aa* promoters, derived from *Bacillus thuringiensis* crystal toxin proteins, are sporulation-specific promoters [37,72]. These promoters are designed to be activated during the sporulation process, controlled by the mother cell-specific σ^E^ and σ^ĸ^, to express the heterologous antigens at the appropriate stage during the spore formation [161]. The modification of the promoter ensures that the expression of the antigens is tightly controlled, being triggered specifically during sporulation when the spores are formed. Once the promoter is activated, the heterologous antigen is expressed in the mother cell and subsequently integrated into the developing spore. Another promoter utilized for the recombinant expression of heterologous proteins is the stress-inducible promoter derived from the *B. subtilis* glucose starvation-inducible (*gsiB*) gene. This promoter was suggested to be induced under stressful conditions such as heat and an acidic environment, under the control of the alternative sigma factor σ^B^ [84,85]. Other promoters include the xylose-inducible promoter, which initiates gene expression with the addition of xylose in the culture medium [99]. Depending on the system used, the recombinant proteins may be displayed on the surface of the spore coat or encapsulated within the spore, where they remain protected from environmental degradation.

The unique ability of ‘naive’ (non-recombinant) spores to simultaneously stimulate immune responses and function as a natural adjuvant provides an additional advantage. Studies have shown that these non-recombinant spores can enhance antibody and T cell responses to co-administered antigens, including mucosal IgA and systemic IgG, without requiring genetic fusion of the antigen to the spore surface [162,163]. Additionally, spores co-administered with antigen have been shown to activate dendritic cells via TLR2- and MyD88-dependent pathways, enhancing antigen presentation and immune priming [34,162]. These findings suggested that fully recombinant spore preparation is not always necessary to elicit potent immune responses. Optimizing dosing regimens and administration routes, nonetheless, remains essential to fully exploit the immunostimulatory potential of *B. subtilis* spores in vaccine development.

Alternatively, heterologous antigens can be non-recombinantly expressed on the outer surface of *B. subtilis* spores by incubating the spores with the antigen under specific conditions that facilitate binding [86]. In this approach, the spores are incubated with the heterologous proteins, allowing the antigens to absorb and stably bind onto the spore surface [86]. After incubation, the spores are purified to remove any unbound antigens. The adsorption of heterologous antigens to the spore surface is primarily mediated by hydrophobic and electrostatic interactions rather than by specific associations with the spore coat proteins [80]. This process is most effective under low-pH conditions, where the electrostatic interactions are enhanced [80]. Adsorption efficiency decreases at higher pH levels, suggesting that pH is critical in optimizing antigen binding to the spore surface. Using this approach, no genetic manipulation is required, thereby addressing concerns related to the use of recombinant microorganisms and their potential release into the environment [80]. Multiple studies have demonstrated the feasibility of this method, with antigens such as the *FliD* protein [77], TTFC [80], B subunit of the heat-labile toxin [86], fusion protein 1 (FP1) [33], *fliC* [93], gag p24 protein [34] and the C-terminal region of the circumsporozoite surface protein [111] successfully adsorbed onto spores, eliciting robust immune responses upon administration.

Both recombinant and non-recombinant approaches offer distinct advantages and disadvantages, depending on the specific goals of vaccine delivery. The recombinant approach provides high precision in gene insertion, allowing extensive genetic modification and customization [159]. This precision enables the rapid development of candidate vaccines, particularly when the genome of emerging pathogens is identified. Recombinant proteins can be expressed with or without genetic fusion to anchor proteins, offering flexibility in design and production processes. Since the spores are synthesized in the cytoplasm, heterologous antigens anchored to the coat proteins do not need to cross the cell membrane. This eliminates the risk of incorrect folding and preserves the antigen’s biological structure, ensuring its functionality [160]. Expressing antigens into the robust spore coat without genetic fusion to the anchor proteins enhances their stability, protecting them during transit through harsh environments, such as the gastrointestinal tract. This stability increases the likelihood that intact antigens will reach the immune system, triggering a strong and targeted immune response [37]. However, recombinant antigen display highly depends on precise genetic engineering, which may increase production complexity. Despite this, the approach offers scalability and uniform antigen expression, making it suitable for large-scale manufacturing.

The non-recombinant approach simplifies the production process by eliminating the need for genetic manipulation. In some studies, this approach has demonstrated the ability to efficiently display more heterologous proteins than recombinant methods [86]. It also offers the flexibility to display multiple antigens simultaneously, as shown by the successful presentation of the cholera toxin B subunit in its pentameric form [164]. However, non-recombinant antigen attachment can be inconsistent, potentially leading to lower throughput. The stability of physically bound antigens may also be compromised in acidic environments, such as the stomach, limiting the feasibility of oral administration [77]. Alternative mucosal delivery routes, such as intranasal administration, may offer better immunostimulatory effects [77]. When delivered intranasally, antigens on the spore surface are directly exposed to the immune system, facilitating the recognition of foreign antigens by immune cells and eliciting robust mucosal immune responses.

## 6. Manufacturing of *B. subtilis* Spore-Based Vaccines

During the COVID-19 pandemic, LMICs encountered delayed supplies and distribution inequities, revealing a critical gap in their ability to respond swiftly to health emergencies. In many Asian countries, the pandemic exposed critical vulnerabilities, including limited in-region vaccine research and development (R&D) and manufacturing capacity, overreliance on imports from high-income producers, disruptions in global supply chains, high prices, and shortages of raw materials, cold chain logistics, and skilled personnel, thus severely constraining timely access to vaccines across the region [165,166].

To address these challenges, the Association of Southeast Asian Nations (ASEAN) initiated the ASEAN Vaccine Security and Self-Reliance (AVSSR) initiative, aimed at ensuring sustainable and equitable access to vaccines for all ASEAN countries, particularly during public health emergencies [166]. The AVSSR initiative focuses on strengthening regional capabilities across the entire vaccine value chain, from R&D, manufacturing, and quality assurance to regulatory harmonization, procurement, and distribution [167]. By fostering collaboration among ASEAN countries and promoting investment in local infrastructure and human capital, the initiative seeks to reduce dependency on external suppliers and enhance the region’s preparedness and resilience against future pandemics. Ultimately, AVSSR aspires to build a more self-reliant ASEAN health ecosystem that upholds regional solidarity and health security [167].

In this context, the *B. subtilis* spore-based vaccine platform presents a compelling opportunity to support the AVSSR vision. Unlike mRNA and other advanced technologies that are often costly and subject to complex intellectual property (IP) restrictions, the *B. subtilis* spore platform offers an affordable, scalable, and heat-stable alternative that is particularly suited for LMICs. Its simplicity in production makes it an ideal candidate for local manufacturing and deployment within ASEAN countries. By channeling investment and collaborative R&D efforts into such low-cost, accessible technologies, ASEAN can accelerate regional vaccine innovation and reduce dependence on the more expensive vaccine delivery platforms.

The manufacturing process for recombinant *B. subtilis* spore-based vaccines is highly scalable (Figure 3). The upstream processing (USP) begins with preparing the seed vials containing recombinant *B. subtilis* engineered to carry the gene of interest. In the event of an emerging epidemic or pandemic caused by Disease X, once its genomic sequence is determined, the gene encoding the target antigen can be rapidly inserted into a pre-existing vaccine vector. This is followed by seed train cultivation in shake flasks to propagate the vegetative *B. subtilis* cells and then expanding into small bioreactors to produce sufficient viable bacterial cells for subsequent stages. In larger bioreactors, bacterial sporulation is induced by cultivating the bacterial cells in a nutrient-deficient medium. The seed train is a major part of the manufacturing process and is often overlooked. The characteristics and viability of the bacterial cells must be preserved during the seed train [29]. The inoculum size plays a crucial role in determining the yield and cost of the production. It was reported that a bigger inoculum size reduces the USP cost and increases the number of batches produced; however, it will require more time to process [29]. Another important aspect of recombinant protein production in bacterial cells is the possibility of loss of plasmid vectors bearing the recombinant gene during bacterial cell division [168]. This may further lead to the production of plasmid-free cells, thus reducing the total yield per batch. The downstream processing (DSP) begins with harvesting the recombinant bacterial spores, usually via centrifugation or filtration, followed by clarification and purification. A common method involves physical or chemical inactivation to ensure sterility and safety. Alternatively, the highly scalable and cost-effective aqueous two-phase systems (ATPSs) can be employed in the purification process [169]. Through ATPS, the spores are separated based on the hydrophobicity, size, or density in a liquid–liquid separation phase, typically polyethylene glycol (PEG) and salt such as NaCl [169]. Following purification, ultrafiltration and diafiltration (UF/DF) are used to concentrate the spores and remove residual impurities, after which a final sterile filtration ensures product safety. The resulting vaccine bulk is then lyophilized to enhance stability during storage and distribution. Depending on the delivery requirements, the end product can be formulated as a solid, such as a tablet or capsule, or maintained as a liquid.

The USP is the initial bioprocessing step and typically requires cell culture to produce antigens (Table 2). Unlike viral vector and inactivated whole-cell vaccines, which often rely on mammalian cell cultures derived from animal sources and necessitate virus seed banks and high-consequence pathogen biocontainment facility and measures to prevent accidental virus release, *B. subtilis* spore-based vaccine production employs basic fermentation techniques of GRAS bacterial cells [170]. This avoids the risks and ethical concerns associated with animal-derived materials while offering a relatively safer manufacturing process due to the low biohazard risk of *B. subtilis* spores. Additionally, unlike mRNA vaccines, which require expensive and intricate in vitro transcription (IVT) enzymatic reactions and stringent cleanroom conditions to prevent RNA degradation, *B. subtilis* spore-based vaccines use inexpensive raw materials to produce Drug Substance (DS) [171]. The DSP of *B. subtilis* spore-based vaccines involves simple centrifugation for the purification and inactivation of the DS, eliminating the need for complex chromatography procedures that are typical of mRNA, viral vector, and protein subunit vaccine production [171,172]. Furthermore, *B. subtilis* spore-based vaccine manufacturing is faster, taking only several weeks compared to the time-consuming processes of inactivated whole-cell vaccines. Its scalability is also a significant advantage, as large quantities of spores can be produced using basic fermentation technology without requiring substantial infrastructural investment. This straightforward approach not only reduces the overall production costs but also makes *B. subtilis* spore-based vaccines an affordable and accessible option for LMICs, particularly when compared to the more demanding production requirements of other vaccine platforms.

**Table 2 vaccines-13-00995-t002:** Comparison of key parameters across different vaccine platforms.

Feature	*Bacillus subtilis* Spore-Based Vaccines	mRNA Vaccines	Viral Vector Vaccines	Inactivated/Attenuated Whole-Cell Vaccines	Protein Subunit Vaccines
**Upstream Process (USP)**	Sporulation of bacterial cells carrying recombinant proteins or adsorption of non-recombinant proteins onto spores	IVT enzymatic synthesis of mRNA with capping and tailing	Cell culture-based production (e.g., HEK293 or CHO cells)	Cell culture (e.g., Vero cells) or embryonated eggs	Varies: Cell culture (e.g., CHO, yeast, or insect cells)
**Downstream Process (DSP)**	Heat-killing or inactivation of spores, followed by centrifugation, UF/DF	Enzymatic digestion, purification using chromatography and UF/DF	Chromatography, UF/DF, and depth filtration	Inactivation, followed by chromatography, UF/DF, and depth filtration	Chromatography, UF/DF, stabilization processes
**Approximate Upscale Volume (Batch)**	High	Moderate	High	High	Moderate
**Time for Development**	Moderate (several weeks)	Rapid (several days to weeks)	Moderate (several weeks)	Moderate to slow (several months)	Moderate (several weeks)
**Scalability**	High	Moderate	High	Moderate	Moderate
**Stability (Cold Chain)**	Room temperature stable	Requires -20 °C to -80 °C	Requires -20 °C to -80 °C	Requires refrigeration (2 °C to 8 °C)	Requires refrigeration (2 °C to 8 °C)
**Cost**	Low	High	Moderate	Moderate	Moderate
**Mucosal Immunity**	Yes	Limited	Limited	Limited	Limited
**Manufacturing Safety Considerations**	Low biohazard risk due to the GRAS status, straightforward handling	Requires stringent cleanroom conditions to prevent RNA degradation	Requires virus seed banks and biosafety measures to prevent virus release	High biocontainment is necessary for infectious viruses; toxic agents for inactivation pose risks	Depending on protein type; moderate biocontainment may be necessary

Bioprocess manufacturing is a fast-growing industry that has evolved significantly over the past two centuries [173]. The future of the industry is likely to be shaped by agile manufacturing, a trend that involves more flexibility, resilience, customization, and affordability in the production processes. One promising advancement is adopting single-use technology/systems (SUT/SUSs), which offer numerous benefits, particularly for spore-based bioprocessing. Adopting the SUT/SUS enhances sustainability by significantly reducing the need for expensive utilities like water for injection (WFI) and steam for cleaning and sterilizing, thereby lowering the risk of contamination and operational cost [174]. It also eliminates the need for extensive cleaning and sterilization validation procedures [174]. Although waste disposal of single-use items might raise environmental concerns, studies suggest that disposing of plastic bags is 50% less energy-intensive than heating of water for sterilization, and it can reduce water consumption in manufacturing facilities by more than 80% [174,175]. Additionally, single-use systems provide greater operational flexibility and require a smaller cleanroom footprint, making them more adaptable to changing manufacturing demands [176,177]. In the context of *B. subtilis* spore-based platforms, SUT/SUSs can be applied throughout both USP and DSP processes. SUT/SUSs are favorable, especially for spore formers, due to the contained use of the upstream process, which reduces the risk of contamination to the facility. Since the removal of spores from equipment is inherently challenging, SUT/SUS provides the optimal advantage in this aspect. In DSP processing, single-use filtration and chromatography cartridges streamline the containment and removal of spores for harvesting, purification, and inactivation.

Another emerging trend is the modular design approach with intensified, continuous biomanufacturing (ICB), which is particularly relevant for spore-based vaccine manufacturing [178]. Modular facilities improve containment with greater flexibility and faster construction timelines while reducing overall costs compared to traditional facilities [178]. These modular systems enable manufacturers to quickly adapt to new production requirements or scale up operations with minimized clean room space utilization, thus reducing footprints [178]. For spore-based vaccine manufacturing, single-use ICB systems address challenges related to CIP and SIP processes, which are critical for spore-forming bacteria. Sporulation induction and harvest can be performed in a continuous flow format; thus, modular ICB facilities can streamline the transition from vegetative growth to spore recovery, reducing batch-to-batch variability. Continuous clarification and separation units are particularly suited for spore suspensions, which are less shear-sensitive compared to mammalian cells or fragile viral particles. This makes *B. subtilis* an ideal candidate for the implementation of intensified continuous processes. Modular biomanufacturing facilities, costing between USD 70 million and under USD 300 million, depending on scale and technology, are considerably more economical and accessible for some LMICs compared to traditional greenfield vaccine facilities, which often require over USD 500 million in upfront capital investment. [174,179,180]. Not only do modular designs substantially reduce capital and construction timelines, but they also better align with LMICs’ infrastructure capacities and regional vaccine needs. Finally, integrating artificial intelligence (AI) into biomanufacturing, as seen with Bio-Processing 4.0, will further revolutionize spore-based vaccine production by enhancing process efficiency, real-time monitoring, and predictive maintenance, paving the way for next-generation bioprocessing [181].

## 7. Challenges and Future Perspectives

While *B. subtilis* spores offer numerous advantages as a vaccine delivery platform, several technical, manufacturing, and regulatory challenges must be addressed to ensure its successful adoption.

From a technical standpoint, studies have highlighted challenges in optimizing the dosage of recombinant *B. subtilis* spores, primarily due to difficulties in accurately quantifying the spores [37,75]. The FDA-approved daily dose of spores per feeding is 1 × 10^9^ CFU/day, which suggests that the actual dose administered to animals may often be much lower than desired [37]. Achieving a sufficiently high dose is crucial to eliciting a robust immune response, particularly when spores are delivered via the oral route, where antigen exposure can be less efficient, especially for the non-recombinant approach [77]. However, concerns have also been raised regarding the potential for immunological tolerance resulting from repeated exposure to high antigen doses from natural exposure to *B. subtilis* and the delivery platform [77,83,88,97]. Nonetheless, some probiotic supplements commonly used in humans are known to deliver doses as high as 100 billion CFU per serving, particularly in formulations designed for gastrointestinal support and immune health. It has been reported in one randomized, double-blind, placebo-controlled study that a daily dose of 100 billion CFU led to better outcomes with reduced incidence of antibiotic-associated diarrhea (AAD) and *C. difficile*-associated diarrhea (CDAD), and fewer gastrointestinal adverse events, compared to 50 billion CFU or placebo [182].

The route of delivery for *B. subtilis* spores presents another technical challenge in optimizing the efficacy of the vaccine. While parenteral routes such as subcutaneous, intraperitoneal, and intramuscular can also be employed, they may not effectively stimulate mucosal immune responses, the initial intent of this platform [78,84]. Mucosal delivery may be more suitable for administering *B. subtilis* spores for it to mount sufficient mucosal immune responses, which is particularly important for protection against respiratory infections [16]. Oral administration, hence, is a preferred mucosal delivery route due to its ability to deliver high antigen doses as well as the ease of administration. The spores’ resistance to harsh gastric conditions enables them to transit through the stomach and reach the intestines intact, where they can interact with the GALT to elicit immune responses [151,152,153,154]. Alternative mucosal delivery routes, such as intranasal, intratracheal, and sublingual administration, present unique opportunities and challenges. While these routes can bypass gastrointestinal barriers and directly target mucosal tissues, they typically involve lower antigen doses [79]. Intranasal delivery has shown promise in inducing localized immunity in the respiratory tract but raises safety concerns, including the risk of organisms crossing the blood–brain barrier, causing neurological side effects [79]. Sublingual and intratracheal delivery, though less studied, offer alternative non-invasive approaches to directly target immune cells in the oral mucosa, but they require further optimization to enhance antigen uptake and immune response [72,101].

From a manufacturing perspective, transitioning from laboratory-scale production to large-scale manufacturing under Good Manufacturing Practices (GMPs) introduces certain levels of complexity [173]. Endospores are widespread in nature and highly resilient, making their containment and eradication a significant challenge. This necessitates stringent manufacturing protocols, which can be daunting for contract manufacturers with limited experience handling spore-forming microorganisms. One of the primary challenges is ensuring effective cleaning and disinfection of the facility, particularly when reusable equipment is involved. The process of killing the spores with the traditional CIP and SIP in a bioreactor poses a big hurdle. The risk of cross-contamination is higher, which requires validated sterilization protocols and comprehensive environmental monitoring. Recognizing these complexities, the US FDA has issued guidance on Manufacturing Biological Intermediates and Biological Drug Substances Using Spore-Forming Microorganisms [183]. This guidance outlines minimum requirements and offers recommendations to help manufacturers maintain compliance. The FDA has revised its earlier stance requiring the use of isolated facilities with permanently designated equipment and materials, allowing for more flexibility to the manufacturers [183]. The advancement of technology in biological manufacturing enables the adoption of more robust validated processes, precise control procedures, and enhanced testing capabilities. The FDA has also recommended that manufacturing facilities for spore-forming organisms be housed in separate, dedicated buildings to minimize the risk of cross-contamination [183]. For facilities producing multiple products, stringent measures are required to ensure containment [183]. This includes the implementation of separate physical containment areas for spore-forming organisms, along with validated monitoring procedures to ensure stringent compliance with containment standards [183].

A critical but often overlooked barrier in vaccine manufacturing, particularly in LMICs, is the availability of reliable water and wastewater infrastructure. Conventional biologics production requires a large volume of purified water, as well as specialized systems to treat and safely discharge effluents generated during fermentation, cleaning, and downstream processing. For LMICs, establishing and maintaining such infrastructure can be cost-prohibitive and logistically challenging. In contrast, spore-based vaccine platforms utilizing *B. subtilis* have comparatively modest requirements. SUT/SUS and ICBs minimize the need for extensive cleaning and water treatment, thereby reducing reliance on centralized water systems. Moreover, the durability of the bacterial spores reduces the demand for stringent water-dependent storage and distribution systems, offering a more sustainable alternative for settings with limited infrastructure.

Additionally, commercializing spore-based vaccines may encounter public perception challenges, such as concerns about safety, environmental release, or doubts over effectiveness, potentially amplifying skepticism. This is especially crucial given the novelty of *B. subtilis* spores as a vaccine platform, which differs from traditional vaccine delivery methods. One major concern lies in the recombinant nature of these spores, which are often classified as Genetically Modified Organisms (GMOs) due to their engineered expression of antigens [184]. This may cause public and regulatory scrutiny, particularly regarding the environmental impact of GMO release. The potential for recombinant spores to germinate in the environment raises questions on unintended ecological consequences [185]. Furthermore, the use of antibiotic resistance genes for the positive selection of the heterologous gene insertion into bacteria poses an additional risk. While the likelihood of gene transfer is low, the possibility of gene dissemination upon environmental release remains a concern [186]. To address these issues, implementing an inactivation procedure could serve as a viable solution. One effective strategy is engineering germination-deficient spores by inactivating genes like *cwlD* and *gerD*, which substantially reduces germination but still elicits immune responses [81]. Another inactivation strategy involves thymineless death, achieved by disrupting the *thyA* and *thyB* genes, making *B. subtilis* spores dependent on external thymine [32,88,184]. Should these spores germinate in an environment where thymine is absent, they undergo immediate cell death, effectively preventing environmental proliferation. Additionally, chemical inactivation such as treating spores with a reagent comprising disodium EDTA, ferric chloride hexahydrate, and ethanol provides an effective way to prevent post-production germination [157,187]. Importantly, inactivation does not seem to impair immunogenicity. In an earlier study, the chemically inactivated *B. subtilis* spores expressing the TonB-dependent receptor (TBDR) of *A. baumannii* were well tolerated and elicited humoral and mucosal immune responses, offering a balanced Th1/Th2 profile while addressing biosafety concerns associated with live recombinant microorganisms [157]. This approach also aligns with public preference for safer, non-viable vaccine platforms while preserving the vaccine’s effectiveness.

Among the most challenging hurdles towards widespread use of spore-based vaccines is regulatory compliance using the current standards set by most regulatory agencies. As a novel platform, these vaccines must undergo comprehensive preclinical and clinical evaluations to address potential concerns regarding their safety, efficacy, stability, biodistribution, immunogenicity, and long-term assessments [188]. This is particularly critical given their innovative nature and unique delivery mechanisms, which may not align with the conventional vaccine categories. This is in spite the fact that *Bacilli*, in general, are non-pathogenic, and *B. subtilis* spores have a well-documented history of safe use, with centuries of consumption as functional foods and dietary supplements in various cultures [145,146,189,190]. This well-established safety profile could provide a strong foundation for regulatory approval, particularly in addressing long-term safety concerns. The safety of spore-based therapeutics has also been demonstrated in humans in a Phase 1b randomized, double-blind, placebo-controlled trial involving 58 patients with mild-to-moderate ulcerative colitis. The spore-based microbiome therapeutic SER-287 exhibited a favorable safety and tolerability profile [191]. However, a key regulatory challenge remains, which lies in determining the most appropriate regulatory classification. Some regulatory agencies might categorize these spore-based vaccines under biologics, given their role in eliciting immune responses and their use in disease prevention. Conversely, others might consider them as functional foods, owing to their historical use in food products and supplements. The lack of a harmonized global regulatory framework adds to this complexity, as different jurisdictions may apply varying criteria for classification and approval [188].

This dual categorization, however, presents opportunities and challenges. On one hand, classifying *B. subtilis* spores as functional foods may streamline regulatory processes in regions with less stringent requirements. On the other hand, biologics classification would necessitate rigorous testing but could enhance their credibility as a validated medical intervention. To navigate this complexity, developers should engage early and proactively with National Regulatory Authorities (NRAs), for example, through preclinical consultations, advisory meetings, and leveraging regional harmonization models such as ASEAN’s GMP mutual recognition. Collaboration with mature regulators or organizations like the WHO, CEPI, and Gavi may help build regulatory capacity and define clear approval pathways tailored to LMICs’ needs. Additionally, regulatory bodies like the FDA could consider licensing the platform as a whole rather than evaluating individual products derived from it.

Given these dual categorization characteristics, establishing a new regulatory category, such as Bioactive Microbial Nutraceutical (BMN), specifically tailored for bioactive microbial delivery systems such as the bacterial spore-based platforms, could offer a more appropriate and effective regulatory pathway. Similarly to existing frameworks used for phytochemicals and nutraceuticals, this category would accommodate health-promoting products that fall outside the conventional definitions of pharmaceuticals or biologics but demonstrate scientifically validated health benefits. The BMN category could encompass bioactive microbial platforms, including *B. subtilis* spores, that serve both immunological and health-supportive functions. By doing so, regulatory authorities would be better equipped to apply context-appropriate safety, quality, and efficacy standards without imposing unnecessarily rigid requirements intended for conventional drug products. This approach would not only accelerate innovation and reduce development costs, especially in resource-limited settings like many ASEAN countries, but would also provide a clear and globally accepted framework for classifying and approving novel, low-cost vaccine technologies. In the long term, the recognition of this new category could catalyze the broader adoption of spore-based vaccines as a credible, accessible solution to strengthen regional and global vaccine self-reliance. Perhaps LMICs such as the ASEAN member states could take the lead in being the first to formally recognize this new category of health products.

## 8. Conclusions

The COVID-19 pandemic exposed significant vulnerabilities in global health security, particularly in LMIC vaccine inequality/equity. Vaccination, however, remains the single most effective global response to emerging infectious disease threats. Despite the remarkable success of mRNA and viral vector vaccines in combating the pandemic, these platforms are associated with significant limitations, including adverse effects, complex manufacturing processes, high costs, and reliance on stringent cold chain logistics. These factors disproportionately impacted LMICs, intensifying the disparities in vaccine distribution. The development of cost-effective and scalable vaccine platforms is, therefore, a critical global health priority, particularly in the preparation against future pandemics. *B. subtilis* spore-based vaccines represent a potential and equitable alternative for vaccine accessibility that could enhance pandemic preparedness. It is particularly an appealing vaccination strategy in LMICs, due to its well-established safety profile, resilience under harsh conditions, and high scalability. In cases of mucosal infections, vaccines administered via mucosal routes are recommended because they have the potential to prevent infection at the point of entry. The heterologous antigens expressed on *B. subtilis* spores have been shown to stimulate both the systemic and mucosal immune responses, thus making them the ideal candidates for use in mucosal vaccination. Considering the benefits it offers, the development of vaccines based on *B. subtilis* spores should be thoroughly considered as a potentially effective tool for resilient health defense, particularly in LMICs.

## Figures and Tables

**Figure 1 vaccines-13-00995-f001:**
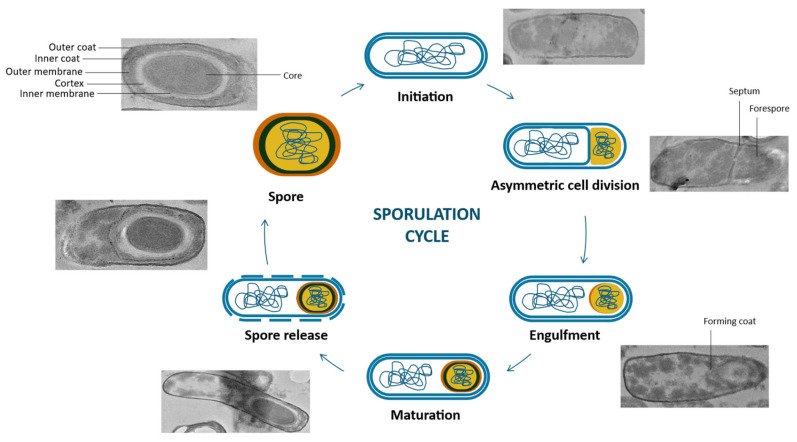
The sporulation cycle of *B. subtilis* illustrated with schematic and transmission electron micrographs. In this process, nutrient deprivation triggers activation of *Spo0A*, initiating sporulation. The cell undergoes asymmetric division, forming a forespore and mother cell; the mother cell engulfs the forespore, which develops a peptidoglycan cortex, while a protective spore coat forms around it. The mature spore accumulates dipicolinic acid and calcium, enhancing its resilience. The mother cell then lyses, releasing a dormant spore capable of germinating under favorable conditions.

**Figure 2 vaccines-13-00995-f002:**
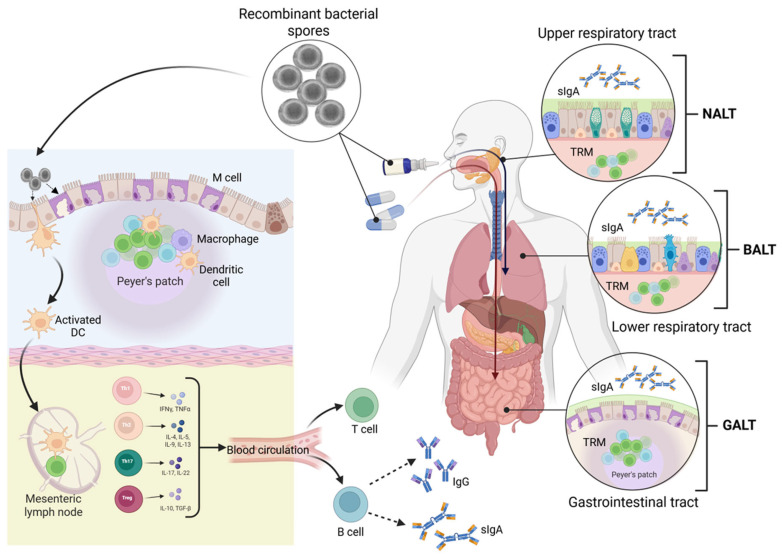
Induction of systemic and mucosal immune responses by recombinant *B. subtilis* spores. The left panel illustrates antigen uptake and initiation of the immune response in the gut. *B. subtilis* spores are sampled by M cells or directly engulfed by dendritic cells (DCs) that extend processes across the epithelium. DCs migrate to mesenteric lymph nodes, where they present spore-associated antigens to naïve CD4^+^ and CD8^+^ T cells. This leads to the differentiation of effector subsets, including Th1, Th2, Th17, and Treg cells. B cells are also activated, undergoing class-switch recombination and differentiating into plasma cells that secrete IgA and IgG antibodies. Activated T cells and antibodies then enter systemic circulation. The right panel shows the distribution of effector responses at mucosal sites. Secretory IgA (sIgA) and tissue-resident memory T cells (TRM) are generated in nasal-associated lymphoid tissue (NALT), bronchus-associated lymphoid tissue (BALT), and gut-associated lymphoid tissue (GALT). These responses provide local mucosal immunity. Created with https://www.biorender.com/.

**Figure 3 vaccines-13-00995-f003:**
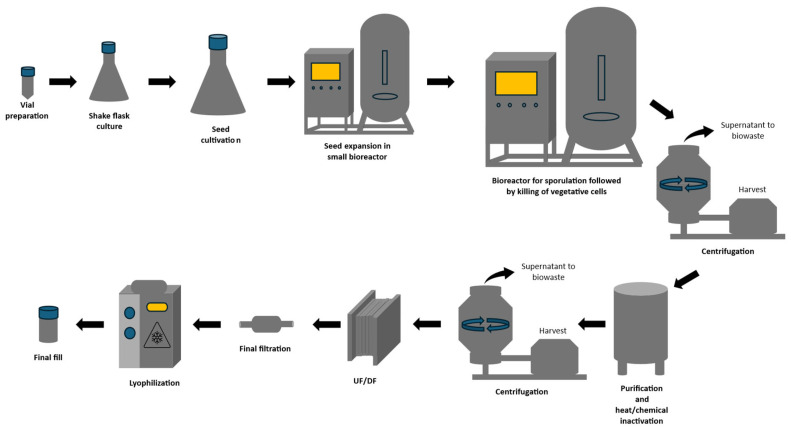
Process flow diagram of *B. subtilis* spore-based vaccine manufacturing. The process begins with vial preparation and seed cultivation in shake flasks, followed by seed expansion in a small bioreactor. Subsequently, the culture is transferred to a larger bioreactor for sporulation, and vegetative cells are killed. The downstream processing involves centrifugation to harvest spores, with the supernatant directed to biowaste. Harvested spores are then subjected to purification and heat/chemical inactivation. Further steps include ultrafiltration/diafiltration (UF/DF), final filtration, lyophilization, and final filling into vials, resulting in the finished vaccine product.

**Table 1 vaccines-13-00995-t001:** Summary of studies utilizing *B. subtilis* spores as a vaccine delivery platform.

Pathogen Targeted	Antigen	Mechanism of Antigen Carrier	Strain	Delivery Route	Application	Refs.
**BACTERIA**						
* **Acinetobacter baumannii** *	TonB-dependent receptors	Recombinant display using a modified cry1Aa promoter	WB800N	Oral	Oral vaccine candidates against *A. baumannii*	[37]
* **Bacillus anthracis** *	Protective antigen (PA)	Recombinant display using a modified cry3Aa promoter	DB104 and WB800N	Oral, Intranasal, Sublingual or Intraperitoneal	Vaccine against *B. anthracis*	[72]
		Recombinant expression using CotB and CotC as anchor motif	PY79	Intraperitoneal		[73]
		Recombinant expression without coat protein	DB104	Intramuscular		[74]
* **Clostridioides difficile** *	C-terminal domain of the spore surface protein BclA3	Recombinant expression using CotB as an anchor motif	PY79	Intranasal	Vaccine against *C. difficile* spores	[75]
	FliD protein fused with human IL-1β domain VQGEESNDK peptide	Recombinant expression using CotG or CotB as anchor proteins	168	Oral	Mucosal immunizations containing *B. subtilis* spores with IL-1β as adjuvant	[76]
	FliD protein	Non-recombinant adsorption onto *B. subtilis* spores with isogenic recombinant BHK121 strain as adjuvant	168 and BHK121	Oral or Intranasal	Mucosal vaccines for *C. difficile* infections	[77]
	Carboxy-terminal repeat domains of toxins A and B	Recombinant expression using CotC and CotB as anchor proteins	PY79	Orogastrically	Vaccine against *C. difficile*	[78]
* **Clostridium tetani** *	Tetanus toxin fragment C (TTFC)	Recombinant expression using CotC anchor motif	Derivatives of strain 168 *trpC2*	Intranasal	*B. subtilis* tetanus vaccines	[79]
		Non-recombinant adsorption onto *B. subtilis* spores	PY79	Intranasal and oral		[80]
		Recombinant expression using CotB as an anchor motif	PY79	Oral and intranasal		[81,82]
			RH103	Oral, intranasal or intraperitoneal		[83]
**Enterohaemorrhagic** ***Escherichia coli*** **(EHEC)**	Shiga-like toxin (Stx)	Recombinant expression induced by the stress-inducible sigma B-dependent promoter derived from the *B. subtilis gsiB* gene	WW02	Oral, intranasal, or subcutaneous	Potential antigen carrier	[84]
**Enterotoxigenic** ***Escherichia coli*** **(ETEC)**	B subunit of the heat-labile toxin (LTB),	Recombinant expression under the control of a stress-inducible promoter derived from the *B. subtilis* glucose starvation-inducible (*gsiB*) gene	WW02	Oral or intraperitoneal	New episomal expression system to improve the performance of *B. subtilis* as a live orally delivered vaccine carrier	[85]
* **Escherichia coli** *	B subunit of the heat-labile toxin	Non-recombinant adsorption onto *B. subtilis* spores	PY79	Intranasal	Mucosal vaccine delivery	[86]
* **Helicobacter acinonychis** *	UreB protein	Recombinant expression using CotC as anchor motif mixed with IL-2-presenting spores (BKH121)	BKH108	Oral	Vaccine candidate supplemented with an appropriate adjuvant	[87]
* **Helicobacter pylori** *	Urease subunit A (UreA) and subunit B (UreB)	Recombinant display of chimeric gene by in-frame fusion to CotB using THY-X-CISE^®^ cloning technique	PY79	Oral	Oral vaccine against *H. pylori*	[88]
	CTB-UreB	Recombinant expression using CotC as an anchor motif	WB600	Oral		[89]
	UreB protein	Recombinant expression using CotC as anchor motif with recombinant spores presenting IL-2 as adjuvant	BKH48 and BKH108	Oral		[90]
* **Mycobacterium tuberculosis** *	Fusion protein FP-1 (Ag85B-Acr-HBHA)	Non-recombinant adsorption onto heat-inactivated *B. subtilis*	HU58	Intranasal	Post-exposure vaccination and booster vaccination	[91]
	Fusion protein 1 (FP1)	Non-recombinant adsorption onto *B. subtilis* spores	-	Intranasal booster	Oral vaccine candidates against *M. tuberculosis*	[33]
	A truncated fusion of Ag85B191-325 and CFP101-70 antigens (T85BCFP)	Recombinant expression using CotC as anchor motif on the spore coat of MTAG1 and in the cytosol of vegetatively grown cells of MTAG2 and MTAG3	PY79	Intranasal		[92]
***Salmonella enterica** * **serovar Typhi**	*fliC*	Non-recombinant adsorption onto *B. subtilis* spores	PY79	Subcutaneous	Delivery of recombinant vaccines against bacterial pathogens	[93]
* **Staphylococcus aureus** *	Mutant staphylococcal enterotoxin B (SEB)	Recombinant expression using CotC as an anchor motif	WB600	Oral	Oral vaccine against *S. aureus*	[94]
* **Streptococcus mutans** *	N-terminal polypeptide of P1 (P1_39–512_)	Recombinant expression without using anchor proteins	WW02	Subcutaneous	Development of anticaries vaccines	[95]
**VIRUS**						
**Coxsackie virus**	VP1	Recombinant expression using CotB as an anchor motif	1A771	Intranasal	CA16 VP1 subunit vaccine	[96]
**Enterovirus 71 (EV71)**	VP1	Recombinant expression CotB as anchor motif	1A771	Oral and Intranasal	Potential vaccine against EV71 infection	[97]
**Foot-and-mouth disease virus**	B subunit of cholera toxin (CT-B) and an epitope box constituted with antigen sites from foot-and-mouth disease virus (FMDV) type Asia 1	Recombinant expression without using anchor proteins	1A751	Oral	*B. subtilis*-based recombinant vaccine for the control and prevention of FMDV infections	[98]
***Human Immunodeficiency Virus*** **(HIV)**	gag p24 protein	Non-recombinant adsorption onto heat-killed *B. subtilis* spores	WW02	Subcutaneous	Immunomodulatory properties of *B. subtilis* spores as adjuvant	[34]
**Human papillomavirus (HPV) type 33**	L1 major capsid protein	Recombinant expression using a xylose-inducible system	KCTC 1326	Subcutaneous	Live or whole-cell vaccines administered by antigen delivery system	[99]
**Group A rotaviruses**	VP6	Recombinant expression through a double-crossover event at the *sacA* locus	168	Intranasal	*B. subtilis* spore-based rotavirus vaccines	[100]
**Influenza A virus (IAV)**	M2e-FP protein (RSM2eFP)	Recombinant display using CotB as an anchor motif	PY79	Aerosolized intratracheal and intragastric	Intratracheal vaccination against H1N1	[101]
	3 molecules of M2e consensus sequence of influenza A viruses, termed RSM2e3			Oral	Recombinant spore-based vaccines against influenza	[102]
	A tandem repeat of 4 consensus sequences coding for human—avian—swine—human M2e (M2eH-A-S-H) peptide	Recombinant expression using either CotB, CotC, CgeA, or CotZ as anchor motifs	BTL 20–BTL 26	Oral	*B. subtilis* spores can serve as antigen carriers and elicit specific immune responses without the need for adjuvants.	[103]
**SARS-CoV-2**	RBD	Recombinant display using CotC as anchor motif	WB800N	Oral	Vaccine-like supplement against respiratory infection	[104]
	RBD and HR1-HR2	Recombinant expression of a chimeric gene by in-frame fusion to CotB or CotC using the THY-X-CISE^®^ cloning technique	PY79	Intranasal	Intranasal booster vaccine against SARS-CoV-2	[32]
**PARASITE**						
* **Clonorchis sinensis** *	CsPmy	Recombinant expression using CotC as anchor motif	WB600	Oral and Intraperitoneal	Potential oral vaccine against *C. sinensis*	[105]
	Enolase from *C. sinensis* (*Cs*ENO)			Oral		[106]
	Eucine aminopeptidase 2 of *C. sinensis* (*Cs*LAP2)					[107]
	*C. sinensis* tegumental protein 22.3 kDa (*Cs*TP22.3)					[108]
	*C. sinensis* TP20.8 (Tegumental Protein 20.8 kDa)					[109]
* **Opisthorchis viverrini** *	Large extracellular loop (LEL) of *O. viverrini* tetraspanin-2 (*Ov*-TSP-2)	Recombinant expression using CotC as anchor motif	WB800N	Oral	Vaccine for control of carcinogenic liver fluke infection in humans	[110]
* **Plasmodium falciparum** *	C-terminal region of the circumsporozoite surface protein	Non-recombinant coupling onto *B. subtilis* spores	KO7	Intranasal	As an adjuvant in a vaccine formulation	[111]
* **Schistosoma japonicum** *	Glutathione *S*-transferase (GST)	Recombinant expression using CotC as anchor motif	WB600	Oral	Potential mucosal delivery vaccination against parasite	[112]

## Data Availability

No new data were created or analyzed in this study.

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
