# Peer review of "Bacillus subtilis Spores as a Vaccine Delivery Platform: A Tool for Resilient Health Defense in Low- and Middle-Income Countries"

_vaccines, 2025, doi:10.3390/vaccines13100995_

Round 1

Reviewer 1 Report

Comments and Suggestions for Authors

The purpose of this manuscript is to describe the use of Bacillus subtilis as a vaccine platform for routine use and pandemic preparedness, specifically in the context of low- and middle-income countries (LMICs). The authors examine vaccine-related challenges faced during the COVID-19 pandemic, describe the B. subtilis vaccine platform, and discuss the potential use cases of this system. The authors conclude that this vaccine platform demonstrates feasibility as an effective, globally deployable strategy.

The topic of this manuscript aligns with the scope of Vaccines and synthesizes current literature. The manuscript is well-written and generally clear. Overall, this is a valuable and timely review that will likely be of benefit to a diverse audience. Major and minor comments follow:

Major:

At times, claims regarding the benefits of the B. subtilis platform may have been overstated. I suggest providing relevant references to back up these claims or softening the language. For example, Lines 145-158: The link between vaccine platform choice and pandemic staffing shortage is tenuous.

Relocating text could enhance readability.

Lines 244-259, 363-378, 405-426: The life cycle of B. subtilis is important but could be condensed and located in one place.

Lines 655-686: This text could be moved to the manufacturing section and should specifically discuss what SUT/SUS and ICBs look like for Bacillus subtilis.

Lines 601-611: This antigen attachment section would fit better in the “antigen attachment to spore” section.

Table 1 is valuable information; however, I couldn’t find a reference to this table in the text. The addition of relevant discussion would be useful.

A deeper discussion of immune responses to the B. subtilis vaccine platform is needed to substantiate conclusions. This could be tied in with a “route of administration” section. Again, protective immunity seems to be a major claim (Lines 783-784) and should be covered in more depth.

There are a few “Error! Reference source not found”: Line 363,522

Minor:

I recommend including a discussion of the vaccine platform earlier in the paper. For example, understanding the role of spore formation and antigen presentation in the introduction before the “Lessons learned from the COVID-19 pandemic” would be helpful for the overall context.

Lines 217-226: I would like to see the inclusion of practical strategies for combating vaccine hesitancy [for this platform].  

A discussion of wastewater or water infrastructure would be beneficial, since this can be a major barrier for any manufacturing facility, especially in LMICs.

Reviewer 2 Report

Comments and Suggestions for Authors

The manuscript by Atiqah Hazan et al. focuses on discussing the possibility of using Bacillus subtilis spore as a Vaccine Delivery Platform. The review is based on the review of many literature sources and will probably be useful for researchers as a systematization of knowledge in this area. However, in its current state, the manuscript requires revision.

  1. It is unclear how the authors selected articles for the review, what the selection was based on, my proposal to include a small section on materials and methods to explain the basis for selecting papers, search words, databases, etc.
  2. The review is too long, contains some irrelevant sections that can be shortened. In particular, Chapter "2. Lessons learned from the COVID-19 pandemic", there is no need to consider the lessons of COVID-19 in such detail, since it is not relevant to the main topic of the review, much less highlight it in a separate chapter. In my opinion, the same information is mentioned in the Introduction section, and this is enough to emphasize the relevance.
  3. Table 1. Opisthorchis viverrini Large extracellular loop (LEL). No information about the protein, only about the domain. Please revise
  4. The manuscript lacks sufficient supporting illustrations and details to make its points clear and convincing. My suggestion it would be better to improve the review, if the authors provided more figures illustrating the immunostimulatory effect of B. subtilis, signaling pathways or etc.
  5. The manuscript lacks more discussion about the advantages and disadvantages of the real results of this approach. In Table 1, the authors only list the papers in which this approach was used, but there is no discussion of how well and effective each case actually prevented infection.

Round 2

Reviewer 2 Report

Comments and Suggestions for Authors

The authors have responded to all my questions therefore I do not have any further comments